# Characteristics and Treatment Strategies for Basicervical and Transcervical Shear Fractures of the Femoral Neck

**DOI:** 10.3390/jcm12227024

**Published:** 2023-11-10

**Authors:** Hiroaki Kijima, Shin Yamada, Tetsuya Kawano, Motoharu Komatsu, Yosuke Iwamoto, Natsuo Konishi, Hitoshi Kubota, Hiroshi Tazawa, Takayuki Tani, Norio Suzuki, Keiji Kamo, Ken Sasaki, Masashi Fujii, Itsuki Nagahata, Takanori Miura, Shun Igarashi, Naohisa Miyakoshi

**Affiliations:** 1Department of Orthopedic Surgery, Akita University Graduate School of Medicine, 1-1-1, Hondo, Akita 010-8543, Japanmiyakosh@doc.med.akita-u.ac.jp (N.M.); 2Akita Hip Research Group, 1-1-1, Hondo, Akita 010-8543, Japan; 3Graduate School of Engineering Science, Akita University, 1-1 Tegatagakuen-machi, Akita 010-8502, Japan

**Keywords:** area classification, bone fixation, finite element analysis, osteosynthesis, femoral neck fracture, basicervical fracture, transcervical shear fracture

## Abstract

This study aimed to define basicervical and transcervical shear fractures using area classification and to determine the optimal osteosynthesis implants for them. The clinical outcomes of 1042 proximal femur fractures were investigated. A model of the proximal femur of a healthy adult was created from computed tomography images, and basicervical and transcervical shear fractures were established in the model. Osteosynthesis models were created using a short femoral nail with a single lag screw or two lag screws and a long femoral nail with a single lag screw or two lag screws. The minimum principal strains of the fracture surfaces were compared when the maximum loads during walking were applied to these models using finite element analysis software. Basicervical fractures accounted for 0.96% of all proximal femur fractures, 67% of which were treated with osteosynthesis; the failure rate was 0%. Transcervical shear fractures accounted for 9.6% of all proximal femur fractures, 24% of which were treated with osteosynthesis; the failure rate was 13%. Finite element analysis showed that transcervical shear fracture has high instability. To perform osteosynthesis, multiple screw insertions into the femoral head and careful postoperative management are required; joint replacement should be considered to achieve early mobility.

## 1. Introduction

In recent years, the treatment methods for femoral neck fractures (AO Classification 31B) and intertrochanteric fractures (AO Classification 31A) have been actively researched, and a consensus has gradually been reached [1,2,3]. 

However, a consensus has yet to be reached regarding the treatment of fractures with a fracture line in the boundary area between these two types of fractures. Moreover, fractures with fracture lines near this boundary are the most difficult to treat, and orthopedic surgeons are constantly conflicted between choosing to perform osteosynthesis or joint replacement surgery. Even if osteosynthesis is chosen, no consensus exists on the type of osteosynthesis implant to be used.

Usually categorized as femoral neck fractures, fractures with a fracture line approximately on the border between the femoral neck fracture (AO Classification 31B) and intertrochanteric fracture (AO Classification 31A) are basicervical fractures (AO Classification 31B3). On the other hand, transcervical shear fractures (AO Classification 31B2.3, so-called Pauwels type III femoral neck fractures) have fracture lines that run from the proximal subcapital to near the distal femoral intertrochanteric region. These are typical cases of fractures with fracture lines near the boundary between femoral neck fractures (AO classification 31B) and intertrochanteric fractures (AO classification 31A).

Basicervical (AO Classification 31B3) and transcervical shear (AO Classification 31B2.3, so-called Pauwels type III femoral neck fractures) fractures have high rotational instability and are difficult to treat due to the shear forces applied to the fracture site, which results in a high proportion of poor outcomes [4,5,6,7,8]. Little research has been conducted to determine the proportion of proximal femoral fractures constituted by basicervical and transcervical shear fractures.

Treatment methods for basicervical fractures (AO Classification 31B3) and transcervical shear fractures (AO Classification 31B2.3, so-called Pauwels type III femoral neck fractures) are still actively debated. However, to the best of our knowledge, the proportion of these two types of fractures among hip fractures has not been investigated outside of our study. Furthermore, there are no single studies comparing these two types of fractures.

At least three papers have been published in 2023 regarding basicervical fractures (AO Classification 31B3). Although they are often classified as AO Classification 31B or femoral neck fractures, basicervical fractures are sometimes treated as extracapsular or trochanteric fractures [9,10,11]. In other words, it is a fracture on the borderline between AO Classification 31A and 31B, and there is no consensus on its classification. There have been several studies regarding stability after osteosynthesis, but no conclusion has been made.

Transcervical shear fractures (AO Classification 31B2.3, so-called Pauwels type III femoral neck fractures) are more debated, and a large number of studies have been published in this regard in recent years [7,8,12,13,14,15,16,17,18,19]. However, almost all research about transcervical shear fractures concerns stability after osteosynthesis. There are no studies comparing them with basicervical fractures, or recommending joint replacement.

This study compared and examined the two types of fractures mentioned above, as fractures near the boundary between femoral neck fractures and trochanteric fractures. Moreover, this study aimed to investigate the frequency, treatment methods, and clinical outcomes of basicervical and transcervical shear fractures by defining them using a comprehensive classification system for proximal femoral fractures that has high reproducibility. This enabled us to clarify the fracture characteristics. We also aimed to compare the fixation properties of each fracture type and each osteosynthesis implant for fractures with a fracture line in the area near the border between femoral neck fractures and trochanteric fractures using finite element analysis.

Therefore, the purpose of this study was to extract basicervical fractures and transcervical shear fractures by only one comprehensive classification and directly compare the frequency and characteristics of these two fracture types using clinical outcomes and finite element analysis in order to provide useful evidence for deciding optimal treatment strategies for each fracture type.

## 2. Materials and Methods

### 2.1. Method 1: Investigating the Frequency and Clinical Outcomes of Basicervical and Transcervical Shear Fractures

This study included all patients with proximal femoral fractures admitted to eight general hospitals within a single prefecture in Japan over a period of 2 years (January 2014 to December 2015). In total, 1042 cases of proximal femoral fractures were included. The mean age of the patients was 82 years (ranging: 26 to 108 years), and there were 209 male and 833 female patients. Approval for this study was granted by the institutional review board of Akita University (IRB No. 2598), and the patients provided informed consent to participate in this study. All patients underwent radiographic and three-dimensional computed tomography to determine the area classification and postoperative clinical outcomes of their treatment strategies. Incomplete proximal femur fractures that can only be detected by magnetic resonance imaging were excluded from this study, and only cases wherein diagnosis was made using X-ray and computed tomography were analyzed. Cases of possible pathological fractures due to tumors were excluded; however, cases of multiple trauma or high-energy trauma due to traffic accidents were not excluded. Imaging data and clinical outcomes were collected by retrospective reference to electronic medical records.

In area classification, the proximal femur is divided into four areas based on three boundaries: the center of the femoral neck, the boundary between the femoral neck and trochanteric region, and a plane connecting the lower edges of the greater and lesser trochanters. Thus, proximal femoral fractures are classified based on the area where the fracture lines occur, such as Types 1, 2, 3, 4, 1–2, 2–3, 3–4, 1–2–3, 2–3–4, and 1–2–3–4 [20,21]. In other words, a basicervical fracture with fracture lines limited to the basal neck area (Area 2) is classified as a Type 2 fracture, while a transcervical shear fracture with fracture lines spanning the subcapital region (Area 1) and basal neck area (Area 2) is classified as a Type 1–2 fracture (Figure 1). X-ray and CT images at the time of injury, postoperative X-ray images, and the clinical outcome at the time of final evaluation were the required data for this study, which were available for all cases, with no missing data. 

There are two major advantages of area classification. First, it is highly reliable. Second, it is extremely simple, and anyone can classify the fracture immediately without looking at a classification diagram, and it has been reported that both its intra- and inter-examiner reliability are higher than those for other classifications [21]. Another is that all hip fractures can be classified. It has been reported that 30% of hip fractures cannot be classified even with the AO classification, which is very comprehensive [20]. Therefore, area classification was used in this study.

Regarding postoperative clinical outcomes, the probability of failure was investigated in each case at the final follow-up. The average follow-up duration was 4 months (1–18 months). Specifically, treatment failure was defined as the occurrence of osteosynthesis implant cut-out or telescoping of the lag screw by 10 mm or more. The failure rate of the different treatment strategies was calculated accordingly.

All necessary images, data, and records were retrospectively reviewed from the patient’s medical records and used in this study. Regarding items other than the ones mentioned above, all treatment options, postsurgical rehabilitation, and postoperative follow-up intervals were determined by the orthopedic surgeon at each facility.

### 2.2. Method 2: Investigating Instability and Fixation Using Osteosynthesis Implants for Basicervical and Transcervical Shear Fractures

A model of the proximal femur was created from the computed tomography images of a typical adult femur (male; age, 31 years; height, 171 cm; weight, 78.5 kg), and typical basicervical and transcervical shear fractures were simulated in the model. Fixation using implants—specifically, a short femoral nail with a single lag screw and a short femoral nail with two lag screws—was performed on the following models: a Gamma 3 nail with a single lag screw (Stryker, Mahwah, NJ, USA) and a Cephalomedullary Asia nail with two lag screws (Zimmer Biomet, Warsaw, IN, USA). Osteosynthesis models were created by fixing each fracture with a long nail in a similar manner (Figure 2). The mechanical Finder (version 11, Standard Edition) (Research Center of Computational Mechanics, Inc., Tokyo, Japan) software was used to evaluate bone strength, construct the finite element model, and perform finite element analysis. A three-dimensional (3D) model of the intact femur was constructed by extracting the region of interest around the cortical bone from each CT scan image of the lower extremity that had not fractured. A similar analysis was previously performed on osteoporotic bone [21]. Details regarding the model construction and the implants, material properties (femur bone and implants), boundary conditions (loads, constraints, interactions), and mesh (femur bone and implants) in finite element analysis were similar to those in that previous study. However, in order to examine the effects of the fracture type in particular, we used the bones of a normal adult man as the subject, with a more rigorous analysis method to determine significance, as described below. 

The subject for the finite element models was a young healthy man (age: 30 years, height: 173 cm, weight: 77.1 kg, mean bone marrow density: 0.724 mg/mm^3^). A CT scan (Revolution CT, GE Healthcare, Wauwatosa, WI, USA) was performed on the subject’s lower extremities based on slice thicknesses of 1.25 mm and 512 × 512 pixels per image. In addition, a bone mass phantom (QRM Quality Assurance in Radiology and Medicine GmbH, Baiersdorfer, Germany) was scanned. Mechanical Finder (version 11, Standard Edition) (Research Center of Computational Mechanics, Inc., Tokyo, Japan) was the software used to evaluate bone strength, as well as to construct the finite element model and perform finite element analysis. A three-dimensional model of the intact femur was constructed by extracting the region of interest around the cortical bone from each CT scan image of the lower extremity that had not fractured. Bone density was assigned to each element based on the Hounsfield unit (HU) value of the CT image, reproducing the subject’s bone structure. The relationship between the HU value and bone density was as follows: $$\rho =\{\begin{array}{c} \;\;\;\;0.0\;\left(\mathrm{HU}<-1\right)\\ 0.88\times \mathrm{HU}-17.1\;\left(\mathrm{HU}=-1\right) \end{array}\;$$

The load profile included the muscle forces at the femur, normalized by the body weight. Each of the values on the load profile was calculated using the musculoskeletal model and validated using in vivo data. To prevent the displacement of the rigid body, the distal end of the femur was fully constrained with six degrees of freedom. The coefficient of friction was 0.3 between the bone and implant, and 0.46 between bones. Meanwhile, the between-implant components were set as tie conditions (bonded to each other).

Nonlinear finite element analysis was performed to include the yield failure of the trabecular bone elements due to compression. The yield strength in compression was based on the yield criterion. The modulus of elasticity after yielding was set to 5%. Young’s modulus of the bone was estimated using Keyak conversion formula. Poisson’s ratio for the bone was set to 0.4. Each implant was made of titanium alloy (Ti-6Al-4V) with Young’s modulus of 113.8 MPa and Poisson’s ratio of 0.34; both properties were homogeneously assigned to each element.

In this study, the minimum principal strain (compressive strain) on the fracture surface was compared among the models under a walking load using the finite element analysis software, Mechanical Finder version 11, Standard Edition (Research Center of Computational Mechanics, Inc., Tokyo, Japan) [21]. The displacement of the rigid body was prevented by completely constraining the distal femur with six degrees of freedom. The bone-to-implant and bone-to-bone friction coefficients were set to 0.3 and 0.46, respectively. Nails, screws, and blade connections were fixed to the osteosynthesis model. A low minimal principal strain indicates a high compressive strain. In other words, a low minimal principal strain indicates an unstable fixation condition. The minimum principal strains of all finite elements of the fracture surface were compared between each model using the student’s *t*-test. A difference in fixation was considered to exist when the *p* value of the minimum principal strain was less than 0.001, and the absolute difference of the minimum principal strain was a clinically meaningful difference greater than 0.00001 using SPSS Statistics version 26 (IBM Corp., Armonk, NY, USA).

## 3. Results

### 3.1. Results 1

Of the 1042 fractures, 10 (0.96%) were classified as Type 2 (basicervical fractures [AO Classification 31B3]; Table 1). Of these Type 2 fractures, one (10%) fracture was treated conservatively; six (67%) were treated with osteosynthesis; and three (33%) were treated with femoral head replacement. Osteosynthesis in all six fractures was performed using short femoral nails with two lag screws, and the failure rate in these cases was 0%.

One hundred (9.6%) of the 1042 proximal femoral fractures were Type 1–2 fractures (transcervical shear fractures [AO Classification 31B2.3]; Table 1). Conservative treatment was performed in three (3%) cases, whereas surgical treatment was performed in the remaining 97 (97%) cases. Among the 97 patients who underwent surgery, 74 (76%) were treated with prosthetic replacement (68 patients underwent femoral head replacement and six underwent total hip replacement). No failures occurred, but one case of dislocation was observed.

Osteosynthesis was selected in the remaining 23 (24%) cases: seven (30%) of these patients were treated with Hanson pins (Stryker, Kalamazoo, MI, USA), six (26%) with compression hip screws, five (19%) with short femoral nails, four (17%) with multiple pinning with cannulated cancellous screws, and one (4%) with long femoral nails. Two lag screws were selected for use in five of the six (83%) cases wherein compression hip screws were used and in three of the five (60%) cases wherein short femoral nails were used. Of the 23 patients who underwent osteosynthesis, one (4.3%) had an implant cut-out and was treated with arthroplasty subsequently, and two (8.7%) had telescoping of 10 mm or more. Thus, the failure rate of osteosynthesis in the Type 1–2 cases was 13%, and all failures occurred in cases of compression hip screw cases with two lag screws.

### 3.2. Results 2

Among the cases wherein an implant was used, the minimum principal strain—indicating fixation stability at the fracture site—was lower in transcervical shear fractures than in basicervical fractures (Figure 3).

When considering the presence of a significant difference in the minimum principal strain between implants (*p* < 0.001) and a clinically meaningful difference of an absolute difference greater than 0.00001, fixation with two lag screws with short or long nails was the most stable option for both basicervical and transcervical shear fractures (Table 2, Table 3, Table 4, Table 5 and Table 6).

## 4. Discussion

In this study, we defined basicervical and transcervical shear fractures using area classification, which is a highly reproducible and comprehensive classification system for proximal femoral fractures. After defining these fractures, we investigated more than 1000 proximal femoral fractures. Only 1% of proximal femoral fractures were basicervical, and osteosynthesis resulted in good clinical outcomes in these cases. However, approximately 10% of the cases involved transcervical shear fractures, indicating that excessive implant telescoping or cut-outs occurred in as many as 10% of the cases after osteosynthesis. The results of the finite element model analysis for the two fracture types showed that transcervical shear fractures were highly unstable. While femoral head or total hip replacement would be effective treatments to facilitate early walking, our analyses clearly showed that if osteosynthesis was to be considered for a transcervical shear fracture, postoperative rehabilitation should be carefully performed using implants with multiple lag screws in the femoral head.

The AO Classification is the most widely used system for classifying proximal femoral fractures. According to this system, femoral neck fractures are classified as 31B fractures and trochanter fractures as 31A fractures; therefore, no consensus has been reached on the treatment strategy for fractures with fracture lines in the border region between 31A and 31B or for those with fracture lines that span both [20,22,23].

Basicervical and transcervical shear fractures are considered to be the most difficult-to-treat fractures in the boundary region between the femoral neck and trochanter. These types of fractures have high rotational instability and are difficult to treat because of the shear force applied to the fracture site; in many cases, the outcomes are poor [4,5,6,7,8]. However, no basic research reports have compared the proportion of these fractures among proximal femoral fractures and examined their degree of instability.

In this study, we investigated all proximal femoral fractures that occurred in a certain geographical region over a certain period. Using area classification, we defined basicervical and transcervical shear fractures that spanned the area from the femoral neck to the trochanter. After classifying the fractures in the study cohort, we investigated the frequency and clinical outcomes of basicervical and transcervical shear fractures. Only 1% of all hip fractures had fracture lines at the exact boundary of AO Classifications 31A and 31B. We conducted finite element analysis, the results of which revealed that basicervical fractures are not highly unstable and can be adequately treated with osteosynthesis using devices to suppress rotational instability, such as two lag screws inserted through the neck into the femoral head.

In contrast, a transcervical shear fracture—in which the medial fracture line of the neck reaches the boundary between 31A and 31B, but the proximal fracture line is subapical—is classified as Type 1–2 using area classification because the fracture line straddles the subapical area (Area 1) to the base of the femoral neck (Area 2). Although transcervical shear fractures occur 10 times more frequently than basicervical fractures, basic research using finite element methodology revealed that this type of fracture is highly unstable and subject to strong shear forces, even when two lag screws are inserted into the femoral head from the neck. The treatment strategy for transcervical shear fractures is controversial, and the actual clinical outcomes of osteosynthesis are poor, as evidenced by the outcomes of the 100 cases in the current study.

This study has some limitations. Patient follow-up was not standardized since the study included different hospitals that followed different treatment plans and postoperative follow-up protocol. In the finite element analysis of this study, a Gamma 3 nail with a single lag screw (Stryker, Mahwah, NJ, USA) and a Cephalomedullary Asia nail with two lag screws (Zimmer Biomet, Warsaw, IN, USA) were used, but neither has not been tested with a similar implant type from another manufacturer. Moreover, in the finite element analysis of this study, there is a lack of past data showing that a statistically significant difference reliably indicates a clinically significant difference in fixation. This study shows that to increase stability when performing the procedure mentioned here for the treatment of transcervical fractures, choosing an implant with a long nail that allows multiple screws to be inserted into the femoral head may be more suitable than other options.

Another limitation of this study is that we were unable to perform finite element analysis on Hanson pins, compression hip screws, and multiple pinning with cannulated cancellous screws. If a manufacturer provides implant data or the implant itself, it will be possible to analyze it using the method of this research in the future; thus, this study will serve as a basis for future research focusing on testing such implants.

In this study, we focused on the fixation of osteosynthesis, so we confirmed the success of the procedure based only on the status of X-ray failure, and at least at the stage when there was no X-ray failure, there was no significantly poor clinical outcome. However, clinical outcomes such as the ability to walk postoperatively and mortality rates have not been confirmed. Although dislocation and infection were not confirmed in cases of basicervical and transcervical shear fractures in which joint replacement was performed, we were unable to follow-up in detail for clinical outcomes other than fixation with osteosynthesis, as described above. Therefore, this is also one of the limitations of this study in addition to the clinical data being old. The reason behind this is that our group had already recognized the instability of transcervical shear fractures long before this study was conducted. Moreover, all the recent data from our group have shown that almost all transcervical shear fractures underwent joint replacement with satisfactory results. Based on the results of this study, many institutions will perform osteosynthesis for basicervical fractures, but will consider joint replacement as the first choice for transcervical shear fractures. It is expected that future reports regarding the clinical outcomes after such a change in policy will be published.

In addition, coronal shear fractures were included in area classification Type 1–2. Type 1–2 fractures are defined as fractures with fracture lines in both Areas 1 (subcapital area) and 2 (base of neck area) [24]. Transcervical shear fractures have fracture lines that exist proximally in Area 1 (subcapital area) and distally in Area 2 (base of the neck area), whereas coronal shear fractures have fracture lines in Area 1 on the anterior surface of the neck and Area 2 on the posterior surface of the neck. This type of fracture is expected to be as unstable as a transcervical shear fracture because of the shear stresses applied; future research should be conducted on coronal shear fractures. Another limitation is that we did not consider the possibility of the late segmental collapse of the femoral head due to osteonecrosis after osteosynthesis [25,26]. In the future, it will be necessary to investigate long-term treatment outcomes and incorporate the occurrence of femoral head necrosis after fracture as an indicator to determine treatment strategy.

Future studies on long-term outcomes based on a more detailed classification and investigation of instability by fracture line location would enable ideal treatment decisions to be made for all proximal femoral fracture types. This study is an important step toward creating an ideal treatment regimen.

## 5. Conclusions

In a study of more than 1000 hip fractures, the incidence of basicervical fractures with a fracture line confined to the basal neck was only 1%, and the same treatment selection—osteosynthesis with implant—as that for other proximal femoral fractures was acceptable. However, transcervical shear fractures, which are shear fractures that occur below the subcapital area and at the basal neck, occur in approximately 10% of cases. According to finite element analysis, transcervical shear fractures are highly unstable and the failure rate of osteosynthesis exceeds 10% in these fractures. Therefore, in cases of transcervical shear fractures, treatment options of femoral head or total hip replacement should be considered to improve early rising from bed and early ambulation.

## Figures and Tables

**Figure 1 jcm-12-07024-f001:**
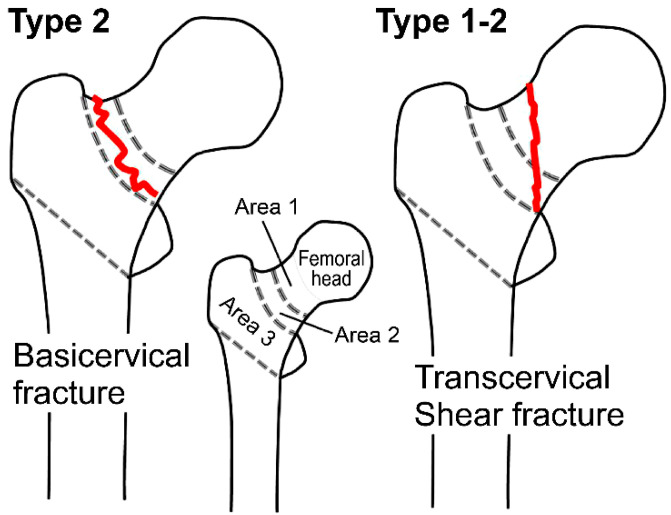
Basicervical fractures (AO Classification 31B3) with fracture lines limited to the basal neck area (Area 2) are classified as Type 2 fractures, while transcervical shear fractures (AO Classification 31B2.3, so-called Pauwels type III femoral neck fractures) with fracture lines spanning the subcapital region (Area 1) and basal neck area (Area 2) are classified as Type 1–2 fractures.

**Figure 2 jcm-12-07024-f002:**
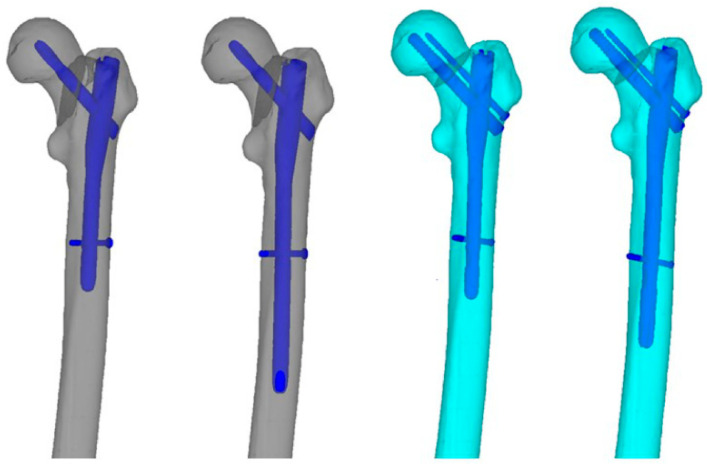
Models were created for basicervical and transcervical shear fractures treated with a short nail with a single lag screw, long nail with a single lag screw, short nail with two lag screws, and long nail with two lag screws.

**Figure 3 jcm-12-07024-f003:**
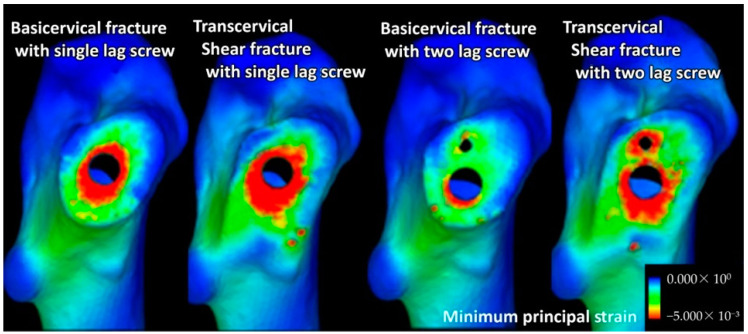
The distribution of the minimum principal strain in the fracture surfaces of basicervical and transcervical shear fractures. Transcervical shear fractures showed a highly compressive strain (red zone; i.e., low minimum principal strain) because of the shearing force at the fracture region.

**Table 1 jcm-12-07024-t001:** The number of basicervical and transcervical shear fractures and the treatment administered.

Treatment	Type 2 (Basicervical) Fracture, n = 10	Type 1–2 (Transcervical Shear) Fracture, n = 100
Cases treated conservatively	1	3
Cases treated with osteosynthesis	6	23
Cases treated with replacement	3	74
Treatment failure (failure rate)	0 (0%)	3 (13%)

**Table 2 jcm-12-07024-t002:** Difference in minimum principal strain between Types 2 and 1–2 fractures.

Implant Combination	Type 2 (Basicervical) Fracture	Type 1–2 (Transcervical Shear) Fracture	*p* Value
Single lag screw Short femoral nail	−0.0008972 ± 0.0011	−0.001258 ± 0.0017	9.2632 × 10^−44^ *
Two lag screws Short femoral nail	−0.0005691 ± 0.00057	−0.001136 ± 0.00095	1.85636 × 10^−26^ *
Single lag screw Long femoral nail	−0.0008033 ± 0.00096	−0.001336 ± 0.0017	1.61333 × 10^−53^ *
Two lag screws Long femoral nail	−0.0005720 ± 0.00057	−0.0009410 ± 0.0010	7.95294 × 10^−69^ *

Minimum principal strain: average ± standard deviation. * *p* < 0.001.

**Table 3 jcm-12-07024-t003:** Difference in minimum principal strain between using single or two lag screws with short femoral nails.

Fracture Type	Single Lag Screw	Two Lag Screws	*p* Value	Absolute Difference
Type 2	−0.0008972 ± 0.0011	−0.0005720 ± 0.00057	6.08949 × 10^−26^ *	0.0002314 **
Type 1–2	−0.001258 ± 0.0017	−0.0009410 ± 0.0010	5.1661 × 10^−104^ *	0.0003901 **

Minimum principal strain: average ± standard deviation. * *p* < 0.001, ** absolute difference greater than 0.00001.

**Table 4 jcm-12-07024-t004:** Difference in minimum principal strain between single and two lag screws with a long femoral nail.

Fracture Type	Single Lag Screw	Two Lag Screw	*p* Value	Absolute Difference
Type 2	−0.0008033 ± 0.00096	−0.0005691 ± 0.00057	6.24132 × 10^−48^ *	0.0003282 **
Type 1–2	−0.001336 ± 0.0017	−0.001136 ± 0.00095	5.0113 × 10^−110^ *	0.0003953 **

Minimum principal strain: average ± standard deviation. * *p* < 0.001, ** absolute difference greater than 0.00001.

**Table 5 jcm-12-07024-t005:** Difference in minimum principal strain between short and long femoral nail with a single lag screw.

Fracture Type	Short Femoral Nail	Long Femoral Nail	*p* Value	Absolute Difference
Type 2	−0.0008972 ± 0.0011	−0.0008033 ± 0.00096	6.24132 × 10^−48^ *	9.38092 × 10^−5^
Type 1–2	−0.001258 ± 0.0017	−0.001336 ± 0.0017	0.000480236 *	7.83294 × 10^−5^

Minimum principal strain: average ± standard deviation. * *p* < 0.001.

**Table 6 jcm-12-07024-t006:** Difference in minimum principal strain between short and long femoral nails with two lag screws.

Fracture Type	Short Femoral Nail	Long Femoral Nail	*p* Value	Absolute Difference
Type 2	−0.0005691 ± 0.00057	−0.0005720 ± 0.00057	0.848366456	−3.02686 × 10^−6^
Type 1–2	−0.001136 ± 0.00095	−0.0009410 ± 0.0010	8.34719 × 10^−9^ *	−7.31347 × 10^−5^

Minimum principal strain: average ± standard deviation. * *p* < 0.001.

## Data Availability

The data that support the findings of this study are available upon request from the corresponding author. The data are not publicly available due to restrictions, e.g., their containing information that could compromise the privacy of research participants.

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
