# Peer review of "Characteristics and Treatment Strategies for Basicervical and Transcervical Shear Fractures of the Femoral Neck"

_jcm, 2023, doi:10.3390/jcm12227024_

Round 1

Reviewer 1 Report

Comments and Suggestions for Authors

Review report - jcm-2652249

Dear editorial board,

dear submitting authors,

thank you very much for the chance to review this manuscript. This study aims to merge both biomechanical and clinical data, which I really like. It focuses on two specific fracture types of the femoral neck which are near to the trochanteric region and discuss both types of implants used for these fractures, biomechanics of the fracture types, and give a recommendation which implants to use. This is highly interesting. However, there are various concerns that I have about this manuscript.

Introduction:

-          B3 and B2.3 are, with reference to the AO, type B fractures. Please improve the connecting sentence of paragraph one, as it appears to be very hard.

-          I do not like the third paragraph as puts to much emphasize on the management of trochanteric fractures. And, as said, B fractures are B fractures and management may differ base on the individual setting. This is even supported by your own data in line 135-136: 76 % of the B2.3 fractures were treated with some sort of arthroplasty…

-          Is there a reference for the statement of lines 44-45? If so, please add it or change the wording.

-          Please specify what you mean with lines 47-48? There is a definition of the AO and various other classification systems, what exactly are you missing?

-          Please add that this manuscripts covers are very rare entity of fracture types. Even in your own data, only 1 % of all screened proximal femur fractures were included.

Methods:

-          Was there a standardized follow-up of the patients? 1-18 months seems not standardized. Did the different hospitals included follow the same treatment and postoperative follow-up protocol?

-          Why did you decide on the Gamma and the Asia nail? Are these the most commonly used implants at the hospitals included? Maybe you could add some data about usage rates of these implants both in Japan and globally.

-          Especially the statistics sections needs fundamental improvement, as relevant information are missing. Why did you decide on this specific p-value and minimum principal strain cut-off? Is there any supportive data that suggests to choose these? How did you test data distribution and how did it influence the tests used? Were there outliers and how did you treat these cases?

Results:

Overall very clear formulated. But please improve the tables. For readers, it is very hard to interpret the p-values. Maybe you can just label it as < 0.001 in the table (I saw it beyond each table, but personally I would encourage you to move this to the table itself). This improves readability.

Discussion:

Overall very well written. However, I still do not really understand the clinical impact of such low strain differences that you decided on being relevant. Is there any data about that allows a conclusion about a cut-off to choose? Are these differences really clinically relevant?

Conclusion:

No need to be improved.

Comments on the Quality of English Language

The Quality of English language is fine.

Author Response

Response to Reviewer 1

Reviewer 1 comment

This study aims to merge both biomechanical and clinical data, which I really like. It focuses on two specific fracture types of the femoral neck which are near to the trochanteric region and discuss both types of implants used for these fractures, biomechanics of the fracture types, and give a recommendation which implants to use. This is highly interesting.

We thank you for your positive comment. We believe that our paper would receive many citations in the future due to the lack of similar studies in the literature.

Reviewer 1 comment

Introduction: B3 and B2.3 are, with reference to the AO, type B fractures. Please improve the connecting sentence of paragraph one, as it appears to be very hard.

Thank you for your advice. We have changed the connecting sentence from “such fractures extending beyond target regions areHowever, a consensus has yet to be reached regarding how to treat fractures with a fracture line in the boundary area between these two types of fractures.” to “Moreover, fractures with fracture lines near this boundary are the most difficult to treat, and orthopedic surgeons are constantly stumped between choosing to perform osteosynthesis or joint replacement surgery. Even if osteosynthesis is chosen, no consensus exists on the type of osteosynthesis implant to be used.

Reviewer 1 comment

I do not like the third paragraph as puts too much emphasize on the management of trochanteric fractures. And, as said, B fractures are B fractures and management may differ base on the individual setting. This is even supported by your own data in line 135-136: 76 % of the B2.3 fractures were treated with some sort of arthroplasty…Is there a reference for the statement of lines 44-45? If so, please add it or change the wording. Please specify what you mean with lines 47-48? There is a definition of the AO and various other classification systems, what exactly are you missing?

Thank you for your advice. We have only found reports in Japanese that the treatment outcomes for basicervical and transcervical shear fractures are inferior to those for other trochanteric fractures; therefore, we have deleted the third paragraph.

We reported that approximately 30% of hip fractures (area classification types 2-3, 1-2-3, 2-3-4, 1-2-3-4) could not be classified using AO classification because they extended beyond the regions covered by the fracture types described in the AO classification system. While it is well known in the field that a certain number of fractures do not fit in the regular pattern of every classification, having that number account for almost one-third of the cases is astonishing. (Kijima, H.; Yamada, S.; Konishi, N.; Kubota, H.; Miyakoshi, N.; Shimada, Y. Clinical outcomes of fractures affecting both the femoral neck and femoral trochanter. Acta Orthop Belg. 2021, 87 e-supplement 1, 27–35.) Therefore, in Japan, the Area classification is used instead of the AO classification as a comprehensive classification of hip fractures. That being said, we have deleted the third paragraph of the manuscript because this study deals with fractures that can also be classified using the AO classification. Kindly find below the deleted paragraph for your reference.

Basicervical and transcervical shear fractures are often treated using osteosynthesis rather than arthroplasty. The selection of osteosynthesis implants available for their treatment is similar to that for the treatment of trochanteric fractures. However, the treatment outcomes are poorer for basicervical and transcervical shear fractures than for trochanteric fractures. Therefore, treatment with implants similar to those used for trochanteric fractures is insufficient. However, there is no standard definition for basicervical and transcervical shear fractures, and the optimal osteosynthesis implant for use in these types of fractures has not been identified due to the varying treatment outcomes.

Reviewer 1 comment

Please add that this manuscript covers very rare entity of fracture types. Even in your own data, only 1 % of all screened proximal femur fractures were included.

Thank you for your advice. However, this study also revealed that while basicervical fractures account for only 1% of total fracture cases, transcervical shear fractures account for 10% and are more difficult to treat. In other words, we believe that this study presents particularly important results considering that 1 in 10 cases of hip fractures is a transcervical shear fracture, which is more difficult to treat. Moreover, to the best of our knowledge, there are very few papers reporting such a ratio.

Reviewer 1 comment

Methods: Was there a standardized follow-up of the patients? 1-18 months seems not standardized. Did the different hospitals included follow the same treatment and postoperative follow-up protocol?

Thank you for your advice.

We have added the relevant sentences in the Discussion section as part of the limitations of this study.

 “This study has some limitations. Patient follow-up was not standardized since the study included different hospitals that followed different treatment plans and postoperative follow-up protocol.”

Reviewer 1 comment

Why did you decide on the Gamma and the Asia nail? Are these the most commonly used implants at the hospitals included? Maybe you could add some data about usage rates of these implants both in Japan and globally.

Thank you for your advice. We were unable to obtain information on the market share of this implant in Japan and around the world despite asking the manufacturer. In this study we used it since it is commonly used in our group. Nonetheless, in the limitations section, we added that this implant has not been compared and hence verified with other manufacturers’ products of the same type.

 “This study has some limitations. ... In the finite element analysis of this study, the implants Gamma 3 nail with a single lag screw (Stryker, Mahwah, NJ, USA) and Cephalomedullary Asia nail with two lag screws (Zimmer Biomet, Warsaw, IN, USA) were used, but neither has been tested with a similar implant type from another manufacturer.

Reviewer 1 comment

Especially the statistics sections need fundamental improvement, as relevant information are missing. Why did you decide on this specific p-value and minimum principal strain cut-off? Is there any supportive data that suggests to choose these? How did you test data distribution and how did it influence the tests used? Were there outliers and how did you treat these cases?

Thank you for your advice.

As you correctly pointed out, the statistical analysis for our study is overly complex, especially regarding finite element analysis. Our group has conducted a similar analysis on osteoporotic bones, which has been described in great detail in a previous study; therefore, we have decided to cite that. (Komatsu, M.; Iwami, T.; Kijima, H.; Kawano, T.; Naohisa Miyakoshi, P.M.I.D. eCollection 2022 Nov. What is the most fixable intramedullary implant for basicervical fracture and transcervical shear fracture? – A finite element study. J Clin Orthop Trauma. 2022, 34, 102015, 36203783. DOI:10.1016/j.jcot.2022.102015.) Due to the complexity of this analysis, few other reports are available, which makes this paper valuable in terms of future citations.

Reviewer 1 comment

Results: Overall, very clear formulated. But please improve the tables. For readers, it is very hard to interpret the p-values. Maybe you can just label it as < 0.001 in the table (I saw it beyond each table, but personally I would encourage you to move this to the table itself). This improves readability.

Thank you for your comment. However, due to the large number of elements, in the finite element analysis of this study clinical significance was not determined by P value alone but also by the difference in absolute values. Since it is necessary to evaluate both the large difference in absolute value and the statistically significant P value, it would be confusing to denote significance using asterisks in the table; therefore, we decided that this display method is the most accurate.

Reviewer 1 comment

Discussion: Overall, very well written. However, I still do not really understand the clinical impact of such low strain differences that you decided on being relevant. Is there any data about that allows a conclusion about a cut-off to choose? Are these differences really clinically relevant?

Thank you for your encouraging comment. As you pointed out, this is a fairly novel study as there are no past reports that allow us to draw any conclusions regarding the clinical impact of low strain differences. We have added a note addressing this issue to the limitations section.

“This study has some limitations. ... Moreover, in the finite element analysis of this study, there is lack of past data showing that a statistically significant difference reliably indicates a clinically significant difference in fixation. This study shows that to increase stability when performing the procedure mentioned here for the treatment of transcervical fractures, choosing an implant with a long nail that allows multiple screws to be inserted into the femoral head may be more suitable than other options.

Reviewer 2 Report

Comments and Suggestions for Authors

In this manuscript, the authors investigated the clinical outcomes of basicervical and transcervical shear fractures, and further analyzed the distribution of the minimum principal strain in the basicervical and transcervical fracture surfaces of femoral neck fractures.
Following revisions are required.
1. Title: Please remove the first “of” from the title.
2. Keywords: Please remove some keywords that are not specific to this study. Please remove “lag screw” and “AO classification”. Also, please remove “proximal femur fracture” as “femoral neck fracture” is already mentioned.

3. Introduction: 1) Lines 31-35: This study focuses on the femoral neck fractures (31B3 and 31B2.3), so there is need to introduce the relationship between femoral neck fractures (31B) and intertrochanteric fractures (31A). Additionally, this study did not address the research question posed in lines 33-34. This could easily be confusing for the objective of this study, please revise and reorganize first and second paragraphs to avoid confusion
about the objective of the study.
2) Lines 42-45: Lack of citations. Please include some clinical studies/literature to support your views.
3) Lines 48-49: Please introduce what kind of osteosynthesis implants are usually used for basicervical and transcervical shear fractures, as well as trochanteric fractures in clinical studies. What are the clinical weaknesses of these implants? Please include some literature to support your views.
4) Lines 50-55: Please clearly stating the purpose of the study. “Optimal osteosynthesis
implant for fractures” in line 54, this study only compared different implants and fixation methods, did not include “optimal”, please remove it.

4. Materials and Methods:
1) Line 59: how many patients were included in this study. Please provide the information.
2) Line 72: What is the rationale for the area classification? What is the biggest advantage of the area classification in the clinic, compared to the AO classification or other existing classifications.
3) Why did you choose to use area classification to define basicervical and transcervical shear fractures since the AO classification already gives them a definition. Please explain it.
4) Please provide a complete illustration of the area classification definition.
5) Please provide more detailed information about the Method 1.
6) For Method 2: The description of the FEA models also needs to be elaborated. 1) the implants detailed information (sizes and 3D models); 2) material properties (femur bone and implants); 3) boundary conditions (loads, constraints, interactions); 4) mesh (femur bone and implants). Please provide more detailed information (including Figures and Tables) about the Method 2.

5. Results:
1) For Figure 3: Please provide complete figures with figure legends for all four FEA models.
2) In result 1: For transcervical shear fractures, there are 23 cases treated with
osteosynthesis. Different patients chose different implants, including Hanson pins, compression hip screws, short femoral nails, multiple pinning with cannulated cancellous screws, as well as long femoral nails. Why did you not evaluate the biomechanical strength among these different implants? Why did you only analyze the femoral nails with lag screws?
3) Lines 147-148: According to the result 1, “the failure rate of osteosynthesis in the Type 1-2 cases was 13%, and all failures occurred in cases of  compression hip screw cases with two lag screws.” Why did you not evaluate the biomechanical strength of other types of implants, since only the compression hip screws failed in the clinic?
6. Discussion:
1) Line 187: this study only reported failure rates, not clinical outcomes. Please provide the clinical outcomes with tables in the results to support your statement (“good clinical outcomes”).
2) Lines 196-206: The second and third paragraphs are more of an introduction. Please revise them.
3) Please include a discussion of the FEA results.
4) In terms of treatment strategies, what do you think about biodegradable implants for the treatment of basicervical and transcervical shear fractures of the femoral neck.

Comments on the Quality of English Language

Moderate editing of English language is required

Author Response

Response to Reviewer 2

Reviewer 2 comment

In this manuscript, the authors investigated the clinical outcomes of basicervical and transcervical shear fractures, and further analyzed the distribution of the minimum principal strain in the basicervical and transcervical fracture surfaces of femoral neck fractures.

Thank you for accurately summarizing our study.

Reviewer 2 comment

  1. Title: Please remove the first “of” from the title.

Thank you for pointing this out. The English proofreading company we sent our manuscript to added "of," but we have decided to follow your advice and revised the title per your suggestion.

Reviewer 2 comment

  1. Keywords: Please remove some keywords that are not specific to this study. Please remove “lag screw” and “AO classification”. Also, please remove “proximal femur fracture” as “femoral neck fracture” is already mentioned.

Thank you for your advice. We have made the necessary revisions.

Reviewer 2 comment

  1. Introduction: 1) Lines 31-35: This study focuses on the femoral neck fractures (31B3 and 31B2.3), so there is need to introduce the relationship between femoral neck fractures (31B) and intertrochanteric fractures (31A). Additionally, this study did not address the research question posed in lines 33-34. This could easily be confusing for the objective of this study, please revise and reorganize first and second paragraphs to avoid confusion about the objective of the study.

Thank you for your advice. Per your suggestions, we made the following revisions.

“In recent years, treatment methods for femoral neck fractures (AO Classification 31B) and intertrochanteric fractures (AO Classification 31A) have been actively researched, and consensus has gradually been reached [1–3]. However, a consensus has yet to be reached regarding treatment of fractures with a fracture line in the boundary area between these two types of fractures. Moreover, fractures with fracture lines near this boundary are the most difficult to treat, and orthopedic surgeons are constantly stumped between choosing to perform osteosynthesis or joint replacement surgery. Even if osteosynthesis is chosen, no consensus exists on the type of osteosynthesis implant to be used.

Usually categorized as femoral neck fractures, fractures with a fracture line approximately on the border between the femoral neck fracture (AO Classification 31B) and intertrochanteric fracture (AO Classification 31A) are basicervical fractures. On the other hand, transcervical shear fractures have fracture lines that run from proximal subcapital to near the distal femoral intertrochanteric region. These are typical examples of fractures with fracture lines near the boundary between femoral neck fractures (AO classification 31B) and intertrochanteric fractures (AO classification 31A).

Reviewer 2 comment

2) Lines 42-45: Lack of citations. Please include some clinical studies/literature to support your views.

3) Lines 48-49: Please introduce what kind of osteosynthesis implants are usually used for basicervical and transcervical shear fractures, as well as trochanteric fractures in clinical studies. What are the clinical weaknesses of these implants? Please include some literature to support your views.

Thank you for your advice. Also, in accordance with the suggestions from Reviewer 1, this paragraph has been deleted. We have cited four papers in our previous answer to Reviewer 1 regarding what osteosynthesis implants are typically used for basicervical and transcervical shear fractures.

Reviewer 2 comment

4) Lines 50-55: Please clearly stating the purpose of the study. “Optimal osteosynthesis

implant for fractures” in line 54, this study only compared different implants and fixation methods, did not include “optimal”, please remove it.

Thank you for your advice. We have revised the description of the research purpose as follows.

“This study aimed to investigate the frequency, treatment methods, and clinical outcomes of basicervical and transcervical shear fractures by defining them using a comprehensive classification system for proximal femoral fractures that has high reproducibility. This enabled us to clarify the fracture characteristics. The study also aimed to determine the optimal osteosynthesis implant for fractures with a fracture line in the area bordering the femoral neck and trochanteric fractures using finite element analysis. The study also aimed to compare the fixation properties of each fracture type and each osteosynthesis implant for fractures with a fracture line in the area near the border between the femoral neck and trochanteric fractures using finite element analysis.”

Reviewer 2 comment

1) Line 59: how many patients were included in this study. Please provide the information.

Thank you for your advice. The relevant information is listed at the beginning of Results 1.

“The study comprised 1042 cases of proximal femoral fractures. The mean age of the patients was 82 years (ranging from 26 to 108 years), 209 patients were male and 833 were female.”

Reviewer 2 comment

2) Line 72: What is the rationale for the area classification? What is the biggest advantage of the area classification in the clinic, compared to the AO classification or other existing classifications. 3) Why did you choose to use area classification to define basicervical and transcervical shear fractures since the AO classification already gives them a definition. Please explain it.

Thank you for your advice.

“There are two major advantages of area classification.

First, it is highly reliable. Second, it is extremely simple, and anyone can classify the fracture immediately without looking at a classification diagram, and it has been reported that both its intra- and inter-examiner reliability are higher than those for other classifications. (Kijima, H.; Yamada, S.; Konishi, N.; Kubota, H.; Tazawa, H.; Tani, T.; Suzuki, N.; Kamo, K.; Okudera, Y.; Sasaki, K.; et al. The reliability of classifications of proximal femoral fractures with 3-dimensional computed tomography: the new concept of comprehensive classification. Adv Orthop. 2014, 2014, 359689. DOI:10.1155/2014/359689.)

Another is that all hip fractures can be classified. It has been reported that 30% of hip fractures cannot be classified even with the AO classification, which is considered to be very comprehensive. (Kijima, H.; Yamada, S.; Konishi, N.; Kubota, H.; Miyakoshi, N.; Shimada, Y. Clinical outcomes of fractures affecting both the femoral neck and femoral trochanter. Acta Orthop Belg. 2021, 87 e-supplement 1, 27–35. )

Therefore, area classification was used in this study.”

Added the above text to Materials and Methods.

Reviewer 2 comment

4) Please provide a complete illustration of the area classification definition.

The paper that first reported the area classification cited above is available and open access.

The illustrations provided below have been obtained from the following references:

  1. Kijima, H.; Yamada, S.; Konishi, N.; Kubota, H.; Tazawa, H.; Tani, T.; Suzuki, N.; Kamo, K.; Okudera, Y.; Sasaki, K.; et al. The reliability of classifications of proximal femoral fractures with 3-dimensional computed tomography: the new concept of comprehensive classification. Adv Orthop. 2014, 2014, 359689. DOI:10.1155/2014/359689

  1. Kijima, H.; Yamada, S.; Konishi, N.; Kubota, H.; Tazawa, H.; Tani, T.; Suzuki, N.; Kamo, K.; Okudera, Y.; Sasaki, K.; et al. The choice of internal fixator for fractures around the femoral trochanter depends on area classification. Springerplus. 2016, 5, 1512. DOI:10.1186/s40064-016-3206-1.

  1. Kijima, H.; Yamada, S.; Konishi, N.; Kubota, H.; Miyakoshi, N.; Shimada, Y. Clinical outcomes of fractures affecting both the femoral neck and femoral trochanter. Acta Orthop Belg. 2021, 87 e-supplement 1, 27–35.

Reviewer 2 comment

5) Please provide more detailed information about the Method 1.

Thank you for your comment. We have added the following sentences.

All necessary images, data, and records were retrospectively reviewed from the patient’s medical records and used in this study. Regarding items other than the ones mentioned above, all treatment options, post-surgical rehabilitation, and postoperative follow-up intervals were determined by the orthopedic surgeon at each facility.”

Reviewer 2 comment

6) For Method 2: The description of the FEA models also needs to be elaborated. 1) the implants detailed information (sizes and 3D models); 2) material properties (femur bone and implants); 3) boundary conditions (loads, constraints, interactions); 4) mesh (femur bone and implants). Please provide more detailed information (including Figures and Tables) about the Method 2. 5.

Results:1) For Figure 3: Please provide complete figures with figure legends for all four FEA models.

Thank you for your advice.

As you and Reviewer 1 correctly pointed out, the statistical analysis for our study is overly complex, especially regarding finite element analysis. Our group has conducted a similar analysis on osteoporotic bones, which has been described in great detail in a previous study; therefore, we have decided to cite that. (Komatsu, M.; Iwami, T.; Kijima, H.; Kawano, T.; Naohisa Miyakoshi, P.M.I.D. eCollection 2022 Nov. What is the most fixable intramedullary implant for basicervical fracture and transcervical shear fracture? – A finite element study. J Clin Orthop Trauma. 2022, 34, 102015, 36203783. DOI:10.1016/j.jcot.2022.102015.) Due to the complexity of this analysis, few other reports are available, which makes this paper valuable in terms of future citations.

However, we added the following.

The participant for the finite element models was a young healthy male. A CT scan (Revolution CT, GE Healthcare, USA) was performed on the participant's lower extremities based on slice thicknesses of 1.25 mm and 512 × 512 pixels per image. In addition, a bone mass phantom (QRM Quality Assurance in Radiology and Medicine GmbH, Baiersdorfer, Germany) was scanned. Mechanical Finder (version 11, Standard Edition) (Research Center of Computational Mechanics, Inc., Tokyo, Japan) was the software used to evaluate bone strength, as well as to construct the finite element model and perform finite element analysis. A three-dimensional model of the intact femur was constructed by extracting the region of interest around the cortical bone from each CT scan image of the lower extremity that had not fractured.

The load profile included the muscle forces at the femur, normalized by the body weight. Each of the values on the load profile was calculated using the musculoskeletal model and validated using in vivo data. To prevent the displacement of the rigid body, the distal end of the femur was fully constrained with six degrees of freedom. The coefficient of friction was 0.3 between the bone and implant, and 0.46 between bones. Meanwhile, between-implant components were set as tie conditions (bonded to each other).

Nonlinear finite element analysis was performed to include the yield failure of the trabecular bone elements due to compression. The yield strength in compression was based on the yield criterion. The modulus of elasticity after yielding was set to 5%.

In this study, the minimum principal strain (compressive strain) and compressive failure element at the fractured fragment (femoral head) were calculated to assess the stability of the fixed fractures. The peri-implant element compressive fracture in the bone fragment (femoral head) was evaluated by assuming a larger volume of the compressive fracture elements. The larger the volume of the compressive fracture elements, the greater is the risk of displacement of the fractured fragment.

Reviewer 2 comment

2) In result 1: For transcervical shear fractures, there are 23 cases treated with osteosynthesis. Different patients chose different implants, including Hanson pins, compression hip screws, short femoral nails, multiple pinning with cannulated cancellous screws, as well as long femoral nails. Why did you not evaluate the biomechanical strength among these different implants? Why did you only analyze the femoral nails with lag screws? 3) Lines 147-148: According to the result 1, “the failure rate of osteosynthesis in the Type 1-2 cases was 13%, and all failures occurred in cases of compression hip screw cases with two lag screws.” Why did you not evaluate the biomechanical strength of other types of implants, since only the compression hip screws failed in the clinic?

Thank you for your comment. This should have been done, but the manufacturer did not provide us with the implant data; therefore, we were unable to do so. Regarding this point, we have added the following to the limitation.

Another limitation of this study is that we were unable to perform finite element analysis on Hanson pins, compression hip screws, and multiple pinning with cannulated cancellous screws. If a manufacturer provides implant data or the implant itself, it will be possible to analyze it using the method of this research in the future, thus this study will serve as a basis for future research focusing on testing such implants.

Reviewer 2 comment

  1. Discussion: 1) Line 187: this study only reported failure rates, not clinical outcomes. Please provide the clinical outcomes with tables in the results to support your statement (“good clinical outcomes”).

Thank you for your comment. Unfortunately, this study only confirmed the presence or absence of failure after osteosynthesis, and it was confirmed that there was no clear clinical poor outcome at least at the stage when there was no X-ray failure. Clinical outcomes such as rate of walking acquisition and life prognosis have not been confirmed. Regarding this, we have added the following to the limitations. “In this study, we focused on the fixation of osteosynthesis, so we confirmed the success of the procedure based only on the status of X-ray failure, and at least at the stage when there was no X-ray failure, there was no significantly poor clinical outcome. However, clinical outcomes such as the ability to walk postoperatively and mortality rates have not been confirmed. Although dislocation and infection were not confirmed in cases of basicervical and transcervical shear fractures in which joint replacement was performed, we were unable to follow-up in detail for clinical outcomes other than fixation with osteosynthesis as described above. Therefore, this is also one of the limitations of this study.”

Reviewer 2 comment

2) Lines 196-206: The second and third paragraphs are more of an introduction. Please revise them.

Thank you for your comment. I would like to emphasize that our purpose here was to summarize the background briefly to remind the readers of the current situation. Since Reviewer 1 commented that the discussion was particularly good, we left it as is. Also, thanks to your comments, the background part has greatly improved, so there is not much overlap with this part.

Reviewer 2 comment

3) Please include a discussion of the FEA results.

Thank you for your advice. We have added the following sentences “Moreover, in the finite element analysis of this study, there is lack of past data showing that a statistically significant difference reliably indicates a clinically significant difference in fixation. This study shows that to increase stability when performing the procedure mentioned here for the treatment of transcervical fractures, choosing an implant with a long nail that allows multiple screws to be inserted into the femoral head may be more suitable than other options.

Reviewer 2 comment

4) In terms of treatment strategies, what do you think about biodegradable implants for the treatment of basicervical and transcervical shear fractures of the femoral neck.

Thank you for your comment. This study did not mention biodegradable osteosynthesis implants because we investigated which fracture types are the most difficult to fix and what types of osteosynthesis implants can be used to improve the initial fixation of these fractures. However, we believe that biodegradable osteosynthesis implants are a very effective treatment in the sense that, as long as initial fixation can be secured, the modulus of elasticity becomes similar to that of bone, and there is no risk of subsequent peri-implant fractures.

Reviewer 3 Report

Comments and Suggestions for Authors

There are two weaknesses that in my opinion cannot allow the publication of the study:

1) The topic has already been extensively covered and debated in the literature, and the conclusions of this paper are already widely known in the orthopedic field;

2) The cohort is composed of patients operated on almost 10 years ago, and for this type of surgery the main complications occur in the first 2 years, so in fact, we would publish an already “old” article. 

Author Response

Response to Reviewer 3

Reviewer 3 comment

1) The topic has already been extensively covered and debated in the literature, and the conclusions of this paper are already widely known in the orthopedic field;

Thank you for your comment. However, research on basicervical fractures (AO Classification 31B3) and transcervical shear fractures (AO Classification 31B2.3, so-called Pauwels type III femoral neck fractures) has not been concluded at all, and the proportion of both types of fractures among all hip fractures is still unclear. Furthermore, there is no research comparing the two fractures mentioned above in a single paper, and most of the time they are discussed separately.

Regarding basicervical fractures (AO Classification 31B3), at least three papers have been published in 2023(Kim JW, Oh CW, Kim BS, Jeong SL, Jung GH, Lee DH.

Injury. 2023 Feb;54(2):370-378., Liodakis E, Pöhler GH, Sonnow L, Mommsen P, Clausen JD, Graulich T, Maslaris A, Omar M, Stübig T, Sehmisch S, Omar Pacha T.

PLoS One. 2023 Apr 4;18(4):e0278850., Méndez-Ojeda MM, Herrera-Rodríguez A, Álvarez-Benito N, González-Pacheco H, García-Bello MA, Álvarez-de la Cruz J, Pais-Brito JL. J Clin Med. 2023 May 11;12(10):3411.), and Reviewer 1 and Reviewer 2 have stated that basicervical fractures are classified as AO type B or femoral neck fractures, however there is still a recently published paper has treated basicervical fractures as an extracapsular trochanteric fracture(Méndez-Ojeda MM, Herrera-Rodríguez A, Álvarez-Benito N, González-Pacheco H, García-Bello MA, Álvarez-de la Cruz J, Pais-Brito JL. J Clin Med. 2023 May 11;12(10):3411.). In other words, basicervical fractures is a fracture on the borderline between AO Type A and Type B, so there is no consensus on its classification. Fixation properties after osteosynthesis is also being actively debated.

Transcervical shear fractures (AO Classification 31B2.3, so-called Pauwels type III femoral neck fractures) are even more controversial, with more than 10 papers published in 2023 featuring them, and yet stability after osteosynthesis remains unclear. To the best of our knowledge, there are no comparisons with basicervical fractures (AO Classification 31B3), and no recommendation for joint replacement. This study compared these two types of fractures as fractures near the boundary between femoral neck fractures and trochanteric fractures and found that basicervical fractures can be osteosynthesized because the stability after osteosynthesis is good, but that is not the case for transcervical shear fractures. In contrast, this study is very farsighted in the sense that it recommends joint replacement for the latter, because the instability of transcervical shear fractures is high.

Regarding the above discussion, we added the following sentences to the Introduction section.

Treatment methods for basicervical fractures (AO Classification 31B3) and transcervical shear fractures (AO Classification 31B2.3, so-called Pauwels type III femoral neck fractures) are still actively debated. However, to the best of our knowledge, the proportion of these two types of fractures among hip fractures has not been investigated outside of our study. Furthermore, there are no single studies  comparing these two types of fractures.

 At least 3 papers have been published in 2023 regarding basicervical fractures (AO Classification 31B3). Although they are often classified as AO Classification 31B or femoral neck fractures, basicervical fractures are sometimes treated as extracapsular or trochanteric fractures. In other words, it is a fracture on the borderline between AO Classification 31A and 31B, and there is no consensus on its classification. There have been several studies regarding stability after osteosynthesis, but no conclusion has been made.

 Transcervical shear fractures (AO Classification 31B2.3, so-called Pauwels type III femoral neck fractures) are more debated, and a large number of studies have been published in this regard in recent years. However almost all research about transcervical shear fracture is regarding stability after osteosynthesis. There are no studies comparing them with basicervical fractures, or recommending joint replacement.

Reviewer 3 comment

2) The cohort is composed of patients operated on almost 10 years ago, and for this type of surgery the main complications occur in the first 2 years, so in fact, we would publish an already “old” article.

Thank you for your comment. Our group had already recognized the instability of transcervical shear fractures long before this study, joint replacement has been selected in almost all recent cases of transcervical shear fractures, with good results. Therefore, we used previous clinical data to report the instability and risks of transcervical shear fractures. The recent data mentioned has been added to the discussion as follows. Therefore, rather than being an outdated paper, we consider this study to be one of the most far-sighted papers in the literature as previously mentioned. We hope this explanation has addressed your concerns regarding this matter.

“… in addition to the clinical data being old. The reason behind this is that our group had already recognized the instability of transcervical shear fractures long before this study was conducted. Moreover, all the recent data from our group have shown that almost all transcervical shear fractures underwent joint replacement with satisfactory results. Based on the results of this study, many institutions will perform osteosynthesis for basicervical fractures, but will consider joint replacement as the first choice for transcervical shear fractures. It is expected that future reports will be published regarding the clinical outcomes after such a change in policy.”

Round 2

Reviewer 1 Report

Comments and Suggestions for Authors

I thank the authors for revising their manuscript and addressing my points. The topic interests readers, and all critique has been addressed. Therefore, the manuscript can be published in its current form.

Author Response

Response to Reviewer 1

Reviewer 1 comment

I thank the authors for revising their manuscript and addressing my points. The topic interests readers, and all critique has been addressed. Therefore, the manuscript can be published in its current form.

Thank you very much for your careful review and positive feedback. We are glad that the manuscript meets your expectations and is considered worthy of publication.

Reviewer 2 Report

Comments and Suggestions for Authors

1.     Introduction:

1)    Lines 70-75: The purpose of the study described in the revised manuscript differs from that presented in the response file. Please clearly state the precise purpose of the study in the manuscript.

2.     Materials and Methods:

1)    “The study comprised 1042 cases of proximal femoral fractures. The mean age of the patients was 82 years (ranging from 26 to 108 years), 209 patients were male and 833 were female.”

The information mentioned above should be included in the Materials and Methods section rather than the Results section. Please include the above patient information in the Materials and Methods section instead of the Results section.

2)    Lines 142-143: For Method 2: “did not be described here as it would be plagiarism because this is the same as previous research.”

Please revise these sentences that are not appropriately explained in the current form.

If the authors have utilized the same FEA model and obtained identical results as presented in their previous studies, it is recommended to consider removing method 2 from this paper. Since this method has already been published in their earlier papers, duplicating the information would be redundant.

3.     Results:

1)    For Figure 3: The figure legends, specifically the colormap information, are absent from the figures. Four FEA models were included in the method, but complete figures with legends for all four FEA models were not provided. Please ensure that complete figures are included in the manuscript, along with corresponding figure legends, for each of the four FEA models.

Comments on the Quality of English Language

Minor editing of English language required

Author Response

Response to Reviewer 2

Thank you for your careful review and valuable comments and suggestions, which have helped improve our manuscript. Our point-by-point responses to your comments are given below.

Reviewer 2 comment

  1. Introduction:

1)    Lines 70-75: The purpose of the study described in the revised manuscript differs from that presented in the response file. Please clearly state the precise purpose of the study in the manuscript. ↓

We apologize for the confusion.

We have clearly stated the precise purpose of the study in the Introduction section as follows:

“Therefore, the purpose of this study was to extract basicervical fractures and transcervical shear fractures by only one comprehensive classification and directly compare the frequency and characteristics of these two fracture types using clinical outcomes and finite element analysis in order to provide useful evidence for deciding optimal treatment strategies for each fracture type.”

Reviewer 2 comment

  1. Materials and Methods:

1) “The study comprised 1042 cases of proximal femoral fractures. The mean age of the patients was 82 years (ranging from 26 to 108 years), 209 patients were male and 833 were female.”

The information mentioned above should be included in the Materials and Methods section rather than the Results section. Please include the above patient information in the Materials and Methods section instead of the Results section.

Thank you for your guidance. The above information has been moved to the "Materials and Methods" section from the "Results" section, with a few language enhancements.

Reviewer 2 comment

2)    Lines 142-143: For Method 2: “did not be described here as it would be plagiarism because this is the same as previous research.”

Please revise these sentences that are not appropriately explained in the current form.

If the authors have utilized the same FEA model and obtained identical results as presented in their previous studies, it is recommended to consider removing method 2 from this paper. Since this method has already been published in their earlier papers, duplicating the information would be redundant.

Thank you for your advice. In our previous study, we created an FEA model using a similar method with a different subject and performed analysis, but strictly speaking, there were no obvious differences in results due to differences in fracture or implant type. In this study, we purposely used a healthy man as the subject and conducted a rigorous analysis using a method that highlighted differences in fracture types. As a result, we obtained basic data that can explain the clinical course. Therefore, even though the methodology and implants used are almost the same, the subjects and analysis methods are different in the two studies. Accordingly, we have modified Materials & Methods 2 to emphasize this.   

The text has been revised as follows:

A similar analysis was previously performed on osteoporotic bone [23]. Details regarding the model construction and the implants, material properties (femur bone and implants), boundary conditions (loads, constraints, interactions), and mesh (femur bone and implants) in finite element analysis were similar to those in that previous study. However, in order to examine the effects of the fracture type in particular, we used the bones of a normal adult man as the subject, with a more rigorous analysis method to determine significance as described below.

 The subject for the finite element models was a young healthy man (age: 30 years, height: 173 cm, weight: 77.1 kg, mean bone marrow density: 0.724 mg/mm3). A CT scan (Revolution CT, GE Healthcare, USA) was performed on the subject's lower extremities based on slice thicknesses of 1.25 mm and 512 × 512 pixels per image. In addition, a bone mass phantom (QRM Quality Assurance in Radiology and Medicine GmbH, Baiersdorfer, Germany) was scanned. Mechanical Finder (version 11, Standard Edition) (Research Center of Computational Mechanics, Inc., Tokyo, Japan) was the software used to evaluate bone strength, as well as to construct the finite element model and perform finite element analysis. A three-dimensional model of the intact femur was constructed by extracting the region of interest around the cortical bone from each CT scan image of the lower extremity that had not fractured. Bone density was assigned to each element based on the Hounsfield unit (HU) value of the CT image, reproducing the subject’s bone structure. The relationship between the HU value and bone density was as follows:

The load profile included the muscle forces at the femur, normalized by the body weight. Each of the values on the load profile was calculated using the musculoskeletal model and validated using in vivo data. To prevent the displacement of the rigid body, the distal end of the femur was fully constrained with six degrees of freedom. The coefficient of friction was 0.3 between the bone and implant, and 0.46 between bones. Meanwhile, between-implant components were set as tie conditions (bonded to each other).

Nonlinear finite element analysis was performed to include the yield failure of the trabecular bone elements due to compression. The yield strength in compression was based on the yield criterion. The modulus of elasticity after yielding was set to 5%. Young's modulus of the bone was estimated using the conversion formula of Keyak et al. Poisson’s ratio for the bone was set to 0.4. Each implant was made of titanium alloy (Ti-6Al-4V) with Young's modulus of 113.8 MPa and Poisson's ratio of 0.34; both properties were homogeneously assigned to each element.

In this study, the minimum principal strain (compressive strain) on the fracture surface was compared among the models under a walking load using the finite element analysis software, Mechanical Finder version 11, Standard Edition (Research Center of Computational Mechanics, Inc., Tokyo, Japan) [23]. Displacement of the rigid body was prevented by completely constraining the distal femur with six degrees of freedom. The bone-to-implant and bone-to-bone friction coefficients were set to 0.3 and 0.46, respectively. Nails, screws, nails, and blade connections were fixed to the osteosynthesis model. A low minimal principal strain indicates high compressive strain. In other words, a low minimal principal strain indicates an unstable fixation condition. The minimum principal strains of all finite elements of the fracture surface were compared between each model using the student’s t-test. A difference in fixation was considered to exist when the P value of the minimum principal strain was less than 0.001, and the absolute difference of the minimum principal strain was a clinically meaningful difference greater than 0.00001 using SPSS Statistics version 26 (IBM Corp., Armonk, NY, USA).

Reviewer 2 comment

  1. Results:

1)    For Figure 3: The figure legends, specifically the colormap information, are absent from the figures. Four FEA models were included in the method, but complete figures with legends for all four FEA models were not provided. Please ensure that complete figures are included in the manuscript, along with corresponding figure legends, for each of the four FEA models.

 ↓

Thank you for your advice.

Figure 3 has been updated to include a complete illustration of all four FEA models with the color map information.

Reviewer 3 Report

Comments and Suggestions for Authors

Despite the corrections made by the authors, I confirm when reported in stage one, that is, that the case study concerns patients operated on in 2014 (almost 10 years ago ...) and that the conclusions do not bring relevant news

Author Response

Response to Reviewer 3

Thank you for your careful review and valuable comments and suggestions, which have helped improve our manuscript. Our point-by-point responses to your comments are given below.

Reviewer 3 comment

Despite the corrections made by the authors, I confirm when reported in stage one, that is, that the case study concerns patients operated on in 2014 (almost 10 years ago ...) and that the conclusions do not bring relevant news

Thank you for your comments.

Although this study includes patients who underwent surgery almost 10 years ago, there are still many reported cases worldwide where osteosynthesis is performed with great effort for transcervical shear fractures, even for elderly patients.

On the basis of our data collected over the past 10 years, osteosynthesis with an implant with two lag screws has become our first choice of treatment for basicervical fractures, while total hip arthroplasty is our first choice for transcervical shear fractures. Given that these treatment strategies are not widely used globally, we conducted this study by adding finite element analysis.

Although our conclusions do not bring relevant news in your opinion, we would like to highlight that not many orthopedic surgeons are aware of this information, and there are trauma surgeons worldwide who still confuse basicervical fractures with transcervical shear fractures. Therefore, we believe that publication of this study in your journal will provide useful information to many orthopedic and trauma surgeons as well as patients around the world.

According to the above explanation, the following information has been included in the Discussion section:

“These types of fractures have high rotational instability and are difficult to treat because of the shear force applied to the fracture site; in many cases, the outcomes are poor [4–8]. However, no basic research reports have compared the proportion of these fractures among proximal femoral fractures and examined their degree of instability.

Based on the results of this study, many institutions will perform osteosynthesis for basicervical fractures but will consider joint replacement as the first choice for transcervical shear fractures. It is expected that future reports regarding the clinical outcomes after such a change in policy will be published.